



# Uniform Blade Pitch Misalignment in Wind Turbines: a learning-based detection and classification approach

Sabrina Milani[1], Jessica Leoni[1], Stefano Cacciola[2], Alessandro Croce[2], and Mara Tanelli[1]

[1]Dipartimento di Elettronica, Informazione e Bioingegneria (DEIB), Politecnico di Milano, Piazza L. Da Vinci 32, Milan, 20133, Italy
[2]Dipartimento di Scienze e Tecnologie Aerospaziali (DAER), Politecnico di Milano, Politecnico di Milano, Via La Masa, 34, Milan 20156, Italy

**Correspondence:** Sabrina Milani (sabrina.milani@polimi.it)

**Abstract.** Maintaining wind turbines in efficient and optimal working conditions is crucial to maximize energy production and reduce unexpected downtime, especially in remote or offshore installations. Pitch misalignment is one of the most common issues affecting wind turbine performance. Our previous studies addressed the automatic detection of such fault using either signals from mechanical moments collected from the fixed and rotating reference frames. Specifically, the introduced approaches involve applying machine learning techniques to ad-hoc designed physics-based indicators, extracted from the mentioned signals, to detect the misalignment and localize the fault. Despite these approaches working effectively in case of both single or multiple blades misaligned simultaneously, conditions in which all blades are misaligned by the same quantity have not been taken into account. Unlike individual blade misalignments, this fault presents unique challenges in its detection due to the symmetrical nature of the fault, which minimizes immediate operational disruptions but gradually impacts turbine performance and energy efficiency. To also account for this condition, in this paper, we present an innovative methodology to identify and classify uniform pitch misalignment across all wind turbine blades. This issue has been scarcely explored in existing literature, leaving a critical gap in the understanding and diagnosis of uniform pitch misalignment. Extensive results conducted with linear and turbulent wind conditions prove the effectiveness of our approach at identifying and quantifying the entity of the misalignment, thus paving the way for more efficient and reliable wind turbine diagnostics.

## 1 Introduction

Persistent vibrations caused by aerodynamic imbalances in wind turbines can have a critical impact on the power production efficiency and mechanical health of these systems, potentially leading to failures in essential components as electronics, sensors, gearboxes and blades. These issues often lead to reduced energy production and require frequent maintenance inspections, which can be costly and challenging. Therefore, research has been conducted to design automatic systems to promote the transition from time-based to condition-based maintenance scheduling, leading to savings and reducing downtimes.





## 1.1 Related Works

Among the several issues, pitch misalignment in wind turbines is one of the most common and severe. This issue has been traditionally studied using two main approaches: physics-based methods and machine learning techniques. Physics-based methods, as explored in works such as Dalsgaard et al. (2009) and Cacciola et al. (2016), rely on physical models of the system to identify anomalies like rotor imbalances. Other model-based approaches, discussed in Desheng et al. (2021); Cacciola et al. (2018); Bertelè et al. (2018); Cacciola and Riboldi (2017), aim to address rotor imbalances by employing load compensation or control strategies. All these methods provide a detailed understanding of system dynamics, but they often require a highly specific model tailored to the turbine, limiting their adaptability to different real systems. Moreover, some of the approaches depend on signals that might be difficult to measure in real-world scenarios, limiting large scale adoption. Last, they frequently struggle with accuracy and reliability in turbulent wind conditions, which are typical during wind turbine operations.

On the other hand, machine learning approaches, such as those presented in Kusiak and Verma (2011), or those presented in Cacciola et al. (2016), which includes deep learning models, provide a more flexible and efficient framework. Despite their generality, they lack in interpretability which may be a drawback in industrial applications, both due to certification requirements and lack of insights for maintenance interventions. In addition, these methods face challenges in precisely localizing anomalies or quantifying their severity.

In our earlier works Milani et al. (2024a) and Milani et al. (2024b), to overcome the limitations and combine the strengths of both approaches, we introduced a novel framework leveraging interpretable physics-based features extracted from mechanical moments signals. This approach effectively detects, classifies and localizes the pitch misalignment focusing on cases where one or multiple blades are misaligned simultaneously. Specifically, feature extraction is performed in real-time, focusing on physics-based frequency and time domain features over a fixed number of rotor revolutions to avoid dependency on rotor speed. In particular, the framework was divided into two stages: the first layer detects the presence of the pitch misalignment anomaly and classifies the fault into low, medium, or high severity. If an anomalous rotor behavior is detected, the second layer performs localization by identifying the affected blade(s). Although the framework shows satisfactory performance even under turbulent wind conditions in classifying and localizing the pitch misalignment on the affected blades, the scenario in which all blades are uniformly misaligned has not been considered in our previous studies. Also, to the best of the authors' knowledge, this issue has not even been explored in the existing literature. Indeed, it represents a challenging scenario due to the symmetrical nature of the fault, which complicates the fault detection. However, identifying this fault is critical as the turbine may not exhibit immediate reduction in power production, thus further masking the presence of the fault but affecting severely the mechanical components of the system over time.

Detecting situations where all blades are uniformly misaligned is practically relevant. Consider the most general case where the three blades have generic pitch offsets $\Delta\beta_1$, $\Delta\beta_2$, and $\Delta\beta_3$, collected in the vector $\mathbf{p} = \{\Delta\beta_1, \Delta\beta_2, \Delta\beta_3\}^T$. Any pitch configuration can be decomposed into two parts:

– a **uniform misalignment** defined as: $\Delta\beta_\mathrm{u} = \frac{\Delta\beta_1 + \Delta\beta_2 + \Delta\beta_3}{3}$, which is common to all blades and produces no aerodynamic imbalance;





– a **residual misalignment** $\mathbf{p}_{\text{res}}$ with zero mean, responsible for the aerodynamic imbalance.

This decomposition is unique: each physical misalignment $\mathbf{p}$ corresponds to exactly one pair $(\Delta\beta_{\text{u}}, \mathbf{p}_{\text{res}})$, and vice versa. Therefore, if uniform misalignment is not observable, the three pitch offsets can only be determined up to an unknown constant offset, limiting the detection of the most general form of aerodynamic imbalance.

## 1.2   Problem Statement

As previously mentioned, in the particular case in which all the blades are uniformly misaligned, the fault has a symmetrical nature. This means that loads and vibrations are evenly distributed on the blades and the tower of the turbine. This particular symmetrical behavior can mask the typical symptoms of misalignment, making it more similar to a nominal behavior. As a consequence it becomes harder for traditional diagnostic methods to detect the fault: indeed, the previous architecture would classify these cases as healthy and non-anomalous ones. To tackle this issue, in this work we aim at identifying this condition

by not considering vibrational behavior only or power production, by moving the focus to the entity of the control action. In fact, collective pitch control has a pivotal role in wind turbine systems as it regulates power output production and ensures optimal operational efficiency. The collective pitch system modulates the pitch blade angles of all blades to ensure that the wind turbine operates at its rated power output, in power region III, namely, above the rated wind speed. Whenever a uniform pitch misalignment is present, the regulator might demand unusual collective pitch adjustments to compensate for the altered

aerodynamics behavior to maintain a balance between aerodynamic forces and generator demands. On the other hand, below the rated speed (i.e., in power region II), the system functions in a working regime where the blades pitch is kept constant and the power output is regulated through torque variations. For this reason, the analysis focuses on two distinct wind regimes, and it is designed to be effective not only in ideal wind conditions, but also in turbulent and more realistic scenarios. In Figure 1 the scheme of the overall hierarchical architecture is presented with an additional layer dedicated to the detection of uniform

misalignment. Accordingly, if non-symmetrical misalignment is detected, then the previous presented methodology is applied so that the fault is classified according to its severity and localized on the related blades by considering features related to the mechanical moment features. On the other hand, if a healthy case is detected, an additional verification step needs to be performed. In fact, as previously mentioned, given the symmetrical characteristics of the fault, uniform misalignments could be mistakenly identified as healthy cases by previous approaches. So this new introduced layer is devoted to further check the

supposed healthy cases to identify eventual uniform misalignment conditions. In the latter case, the degree of the misalignment is precisely quantified based on the behavior of the control action in power region III or or the by assessing mechanical moments in within the blades reference frame in power region II.

## 2   Preliminary Analysis

The dataset, consists of several 600-second simulations, generated by a virtual model of a reference 5 MW wind turbine

Jonkman et al. (2009) implemented in the software `Cp-Lambda` (a Code for Performance, Loads, Aeroelasticity by Multi-Body Dynamics Analysis), a state-of-the-art general-purpose multibody simulator Bottasso and Croce (2009–2018). The model





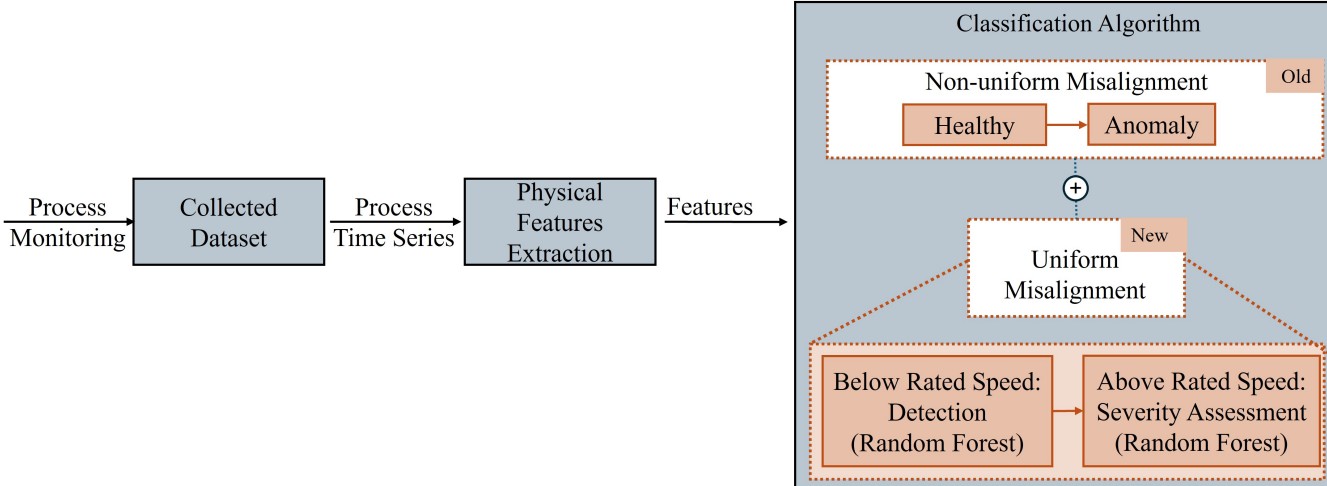

**Figure 1.** Hierarchical Detection and Localization Architecture

.

of the turbine is characterized by flexible tower, blades, and shaft, whereas the blade element momentum theory is used for modeling rotor aerodynamics, including hub- and tip-losses and tower shadow. To measure relevant signals, the turbine model is also equipped with virtual sensors and all simulations were conducted in different pitch misalignment conditions, uniformly

imposed on the three blades. Also, two scenarios have been considered, including turbulent inflow that was defined according to the Standards IEC (2004) (NTM Normal Turbulence Model) and leveraging Normal Weather Predictions (NWP). In addition, the simulations are conducted with an air density set at $\rho = 1.225 \ kg/m^3$ and, and wind speeds span ranging from $v = 5 \ m/s$ to $v = 25 \ m/s$, with the wind direction aligned with respect to the rotor, meaning with zero mean yaw misalignment. The full list of simulations (from cut-in to cut-out speed), was repeated 22 times, with different uniform pitch misalignments values, and

changing every time the turbulence seed for NTM cases. In Table 1, an overview of the key characteristics for the conducted simulations is summarized including information on air density, turbulence level, and wind speed. More in detail, 16 sets of

| Simulations Info | |
|---|---|
| **Simulation Settings** | **Value** |
| Air density | $\rho = 1.225 \ \text{kg/m}^3$ |
| Turbulence Type | NTM – NWP |
| Turbulence Class | C – [–] |
| Wind Speed Range | $v = 5 : 25$ m/s |
| Mean Yaw Misalignment | 0 deg |

**Table 1.** Simulation Parameters





simulations were dedicated to all uniformly misaligned blades and 6 sets for the balanced scenario, *i.e.,* without misalignment. The maximum entity for the considered pitch offset is $\pm 2.0$ deg, while the minimum one is equal to $\pm 0.5$ deg.

## 3  Preliminary Analysis and Features Engineering

The general idea is to compare the behavior of the system in healthy and uniform scenarios and then derive a comprehensive physical understanding of the key differences to define the set of relevant signals from which extract features to quantify the fault when present.

### 3.1  Signals Processing and Exploratory Analysis

According to the physical knowledge of the system, in a balanced rotor, loads are transmitted to the fixed frame at harmonics
multiple of the number $N_B$ of blades only (in the case under study, $N_B = 3$). Conversely, unbalanced rotors transmit loads at all harmonics, with the $1 \times Rev$ frequency being the most evident one.

Therefore, in our previous studies Milani et al. (2024a) and Milani et al. (2024b) the major indicators highlighting the existence of the $1 \times Rev$ harmonic contribution in a non-uniform misaligned scenario were the Mechanical Moments along the lateral and vertical axes. Yet, when all the blades are uniformly misaligned, loads are still transmitted to the fixed frame at harmonics
multiple of the number $N_B$ of blades only, without showing the $1 \times Rev$ frequency component. Therefore, from this first analysis, a healthy case or a uniform misaligned case would be challenging to distinguish as they behave similarly. Thus, according to our previous approaches a uniform misaligned case would be detected as healthy.

Figure 2 illustrates azimuth-based power spectra at a medium wind speed of $v = 15 m/s$ for the Yawing Moment along z direction in a healthy case and different anomalous cases (from $\pm 0.5$ deg and $\pm 2.0$ deg misalignment either in NTM and NWP
scenarios). As in the previous studies, to avoid dependency on the wind speeds, the Fourier Transformations applied on the signals are performed with respect to the Azimuth signal, rather than to time. In such a way, the harmonic response depends on the rotor frequency revolutions only. As previously mentioned, all cases *do* not exhibit a peak placed around the $1\times$ Rev harmonic irrespective of the turbulence condition or the wind speeds.

It is evident that, from the traditional analysis based on the presence of the peak at the 1xRev performed on the spectra, healthy
and anomalous cases cannot be easily distinguished. Therefore, further investigation of additional signal behaviors is necessary. One way to investigate the presence of the fault could be the analysis of the output power production, which indeed may lead to worthless insights, as these systems are controlled by a pitch regulator especially at high wind speeds. Figure 3 shows the power curve in a healthy and faulty case for different wind speeds. As reported in the figure, due to the controller action, especially at high wind speeds, the power curve in a healthy and faulty case are not distinguishable. On the other hand, we rely
on the domain knowledge that whenever multiple blades are affected by pitch misalignment, there may be a noticeable impact on collective pitch demand from the control system. Thus, we investigate if this deviation in the regulator's collective pitch signal can provide a supplementary indicator for detecting and classifying the misalignment.





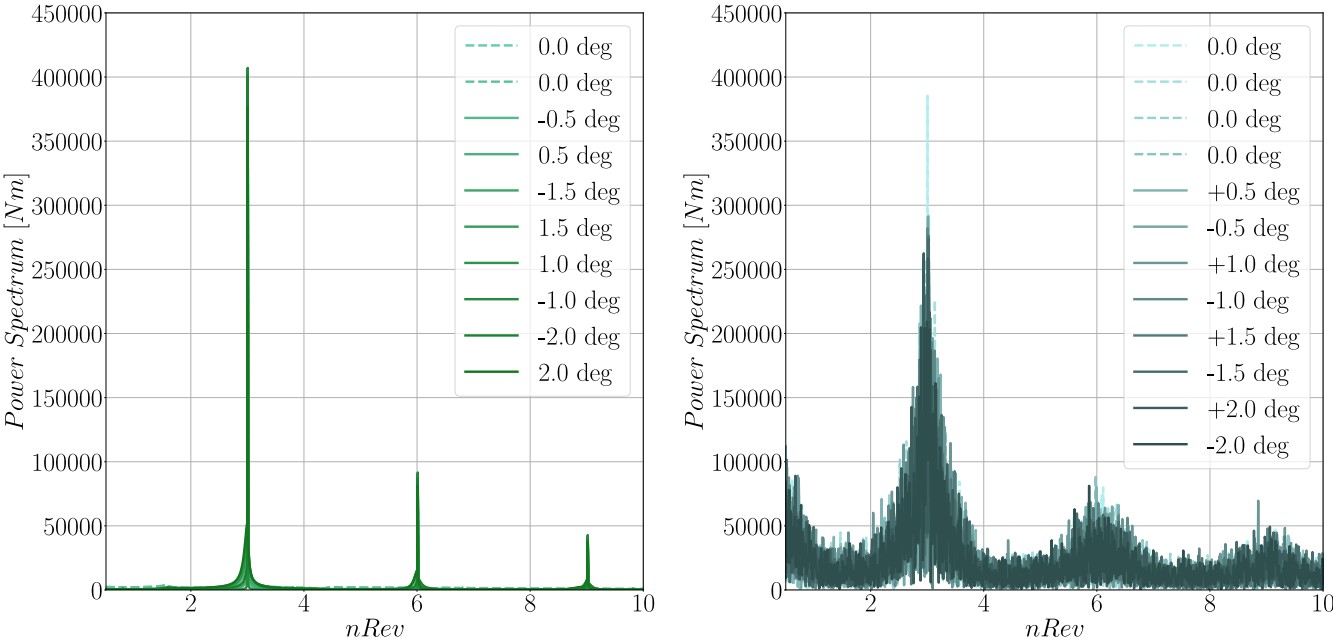

**Figure 2.** Mechanical moments spectra for healthy and different misaligned cases for NWP (on the left) and NTM scenarios (on the right) all at a wind speed of 15 m/s.

.

As expected, we noticed that the pitch adjustments required by the controller in case of misaligned cases might be relevant when compared with respect to the healthy cases and reveal patterns that are distinct from a healthy rotor. Therefore, some

130 relevant features can be extracted fro this signal, such as the mean pitch angle, variance, and the difference of the absolute value of the collective demanded pitch with respect to a healthy case. Additionally, power production might be informative at least until the control action reach the steady state. On the other hand, for wind speeds below the rated value, the pitch controller is not active, as the torque control is responsible for regulating the turbine's operation. Thus, control action-related features are not useful in this region, but mechanical moments measured along the y and z directions on the local reference frame of

135 the blades turned out to be effective as exhibit noticeable deviations when compared to a healthy baseline in the time domain, making them valuable indicators for the analysis in this region.

### 3.2 Features Extraction

In our previous research, we outlined the need for an interpretable machine learning framework for the detection and classification architecture for both certification and adoption purposes. Thus, we take into account interpretability in the whole

design of the presented framework. To this end, we focused solely on extracting physics-based features, which would lead to a better understanding of the decision-making process adopted by the classifier. Also, we choose to rely on an interpretable machine-learning technique, i.e., Random Forest, as capable of explaining their decision making process to the domain expert.

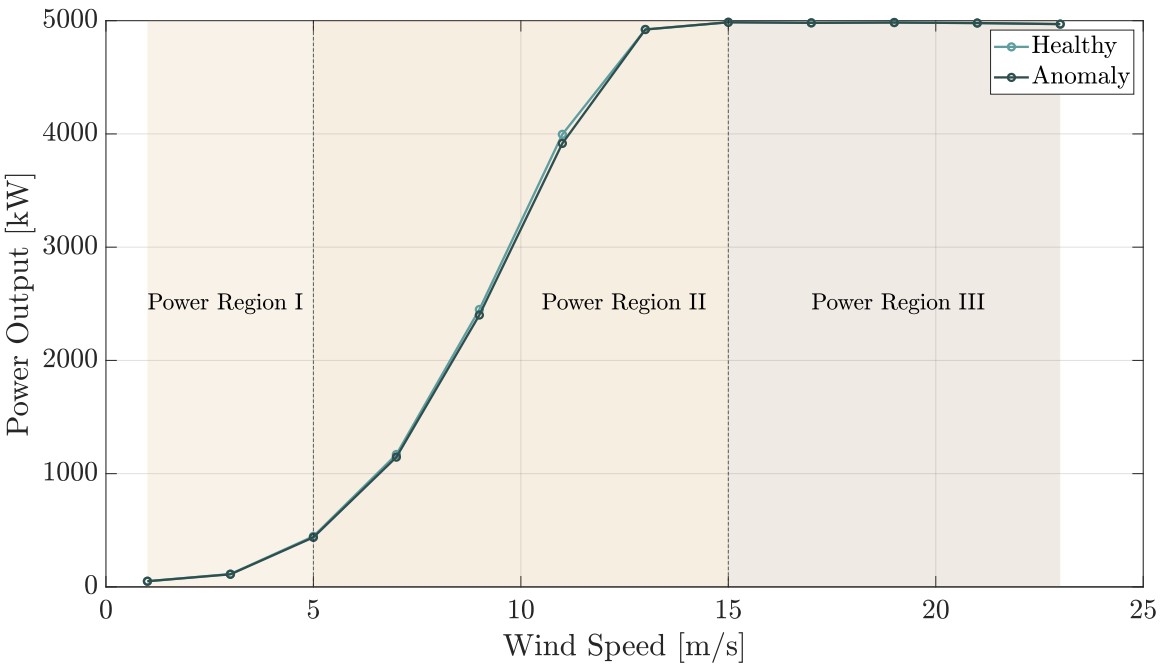

**Figure 3.** Mean Power curves in two different cases: healthy and anomalous

.

According to the preliminary analysis results, given that $1 \times$ Rev harmonics and power curve do not play a crucial role both in detecting the presence of the anomaly, we have focused on extracting features related to the collective demanded pitch and the mechanical moments on the blades in the time domain. Specifically for wind speed higher than the rated speed we extracted:

– $M_p$: mean value in a specified time interval of the collective demanded pitch

– $\sigma_p$: standard deviation value in a time interval of the collective demanded pitch

– $\Delta_p$: the absolute value of the difference between mean collective demanded pitch in a healthy and the considered scenario $\Delta_p = |M_{p,healthy} - M_{p,case}|$. In this case, null values are expected when the current considered case is a healthy scenario, while a difference linearly dependent on the severity of the misalignment is expected when considering an anomalous case

On the other hand, for wind speed lower than the rated speed, we rely on the mechanical moments along y and z direction, and in details we extracted:

– $M_{yi}$ and $M_{zi}$: mean value in a specified time interval of the mechanical moments along y and z on the i-th blade (where $i = 1,2,3$)



– $\Delta_{My}$ and $\Delta_{Mz}$: the absolute value of the difference between mean mechanical moments along y or z in a healthy and the considered scenario $\Delta_{My} = |\sum_{i=1}^{3} M_{y,i} - \sum_{i=1}^{3} M_{y,i}^{H}|$ and $\Delta_{Mz} = |\sum_{i=1}^{3} M_{z,i} - \sum_{i=1}^{3} M_{z,i}^{H}|$ respectively.

These time-domain features are extracted using a moving window approach, as in the original framework, to ensure that the classification algorithm's predictions are generated in real-time. As it is known in the literature, the size of the window plays a prominent role in determining the performance of the detection approach. Thus, we conduct a tailored sensitivity analysis to find the optimal length, as described in Section 4.

## 4 Detection, Severity Assessment and Method

After feature extraction is completed, depending on the operational conditions dictated by the wind speed, two distinct methods are applied. If the working condition is referring to a wind speed lower than the rated speed (i.e., power region II) than the focus is on the mechanical moments on the blade and the coordinates of each feature point are determined by the $\Delta_{My}$ and $\Delta_{Mz}$. In this case, each extracted window corresponds to a point in a 2D space. Conversely, when the wind speed is above the rated speed, then the coordinates of each feature point are determined by the mean or standard deviation of the collective demanded pitch and the difference with respect to a healthy case. Thus, each window corresponds to a point in a 3D space. All these instances derived from this process are exploited for training and evaluating the performance of two different *Random Forest Classifiers*. These classifiers are capable of distinguishing the presence of misalignment from a healthy case, and in case of a uniformly-misaligned rotor by quantifying the severity of the misalignment located on all the blades in power region III. As previously mentioned, the Random Forest algorithm has been selected for its interpretability and robustness, as it is one the most explainable state-of-the-art bagged tree-based classification methods Breiman (2001). This approach combines the outcomes of numerous binary decision trees, which split the data and iteratively,l to reduce variations within the identified nodes at each step. The splitting process is carried out considering specific criteria, including accuracy, precision, and the Gini index Ceriani and Verme (2012). This last metric can be also considered to interpret the classifier logic by quantifying the relevance of each feature in contributing at classifying the instances. In more detail, each classifier (for low and high speeds) was trained on the related instances to recognize from the features of interest in that speed condition the presence of misalignment. The classifier for lower wind speeds is binary, while the other recognizes five conditions: *0 deg* for healthy cases, and for misaligned cases the following misalignment degree $\pm 0.5 deg$, $\pm 1.0 deg$, $\pm 1.5 deg$, $\pm 2.0 deg$. Both classifiers, also work in NTM and NWP conditions. Last, it is important to notice that in designing the architecture, the proper fine-tuning of two sets of hyper-parameters was key, which were the window size for features extraction and the classifiers depth and number of estimators. We adopted a trade off between accuracy and computational time, which increases with the depth of each instance of a tree. Data is split according to a balanced fashion into 70.0% for training and 30.0% for testing in a stratified manner; additionally, 10-fold cross-validation is applied, and the average performance across all folds is reported. Evaluation metrics, including precision, recall, F1-score were chosen for their relevance in evaluating the performance of anomaly detection methods. Precision measures the accuracy of predictions by evaluating the proportion of correctly predicted instances to the total predicted instances for a specific class. In contrast, recall assesses the accuracy of predictions for a class





relative to all actual instances of that class. The F1-score provides a balanced measure by combining both precision and recall,
representing their harmonic mean. Lastly, support represents the number of occurrences of each label within the analyzed class.

## 5   Results and Discussion

This Section reports and discusses the results obtained in the extensive validation of the approach. As reported in Section 1, a main contribution of this work is to address uniformly-distributed misalignment not only in stationary but also turbulent wind conditions. Therefore, we present the results achieved in both stationary and turbulent wind conditions. Fine-tuning of the
parameters shows that the second scenario is more challenging, yet optimal performance can still be achieved.

### 5.1   NWP data

We first evaluate the method considering ideal NWP cases in which no turbulence is considered, and therefore no noise on the signals is present. In this scenario, the classification and detection procedures prove to be highly effective and accurate.

#### 5.1.1   Region II - below rated speed

First, the scenario in which the wind speeds is below the rated wind speed $V = 13m/s$ is considered. As reported in Section 4, the RF classifier dedicated to low speeds performs binary classification (healthy vs. uniform). Therefore, $\Delta_{My}$ and $\Delta_{Mz}$ features are taken into account. As previously mentioned, the moving window length has been carefully chosen to balance resolution on the signals and computational efficiency and a window of 4.3 minutes is chosen in this case, as the minimum value that lead the approach to reach an F1-score of $98.89\%$ in detecting the presence of the anomaly with a wind speed ranging
from $5$ to $11m/s$. In this specific case, the hyperparameter of the model include a number of estimators equal to 2 with depth 5. To provide further insights into the classification outcomes, Figure 4 provide a graphical representation of the results, showing the actual and the estimated instances for each predicted window. In the plot, each point corresponds to a window, and its coordinates are the $\Delta_{My}$ and the $\Delta_{Mz}$. As demonstrated in the Figure, instances are graphically well separable and show a linear trend, allowing the classifier to achieve high classification performance.

#### 5.1.2   Region III - above rated speed

When the wind speed is above the rated speed, we rely on the demanded collective pitch and on the other hand, further classification of the severity of the misalignment can be performed. Average results for evaluation metrics, including precision, recall, F1-Score, and support for this scenario are reported in Tab.2. By considering F1-Score as a metric, our approach proved to be effective at not only detecting the presence of a uniform misalignment but further classifying its severity class, with an
average F1-Score of $97.41\%$ after 4.3 minutes interval and with a wind speed ranging from $13$ to $25m/s$. Here, the model' architecture is slightly more complex, comprising a total of 5 estimators, each with a depth of 20. More specifically, our framework exhibits an excellent F1-Score, achieving 100.0% accuracy in detecting the healthy class, 100.0% for the 0.5 deg 96.0% for the 1.0 deg and 2.0 deg and 95.0% for the 1.5 deg class. Indeed, these achievements stand out as remarkable, since



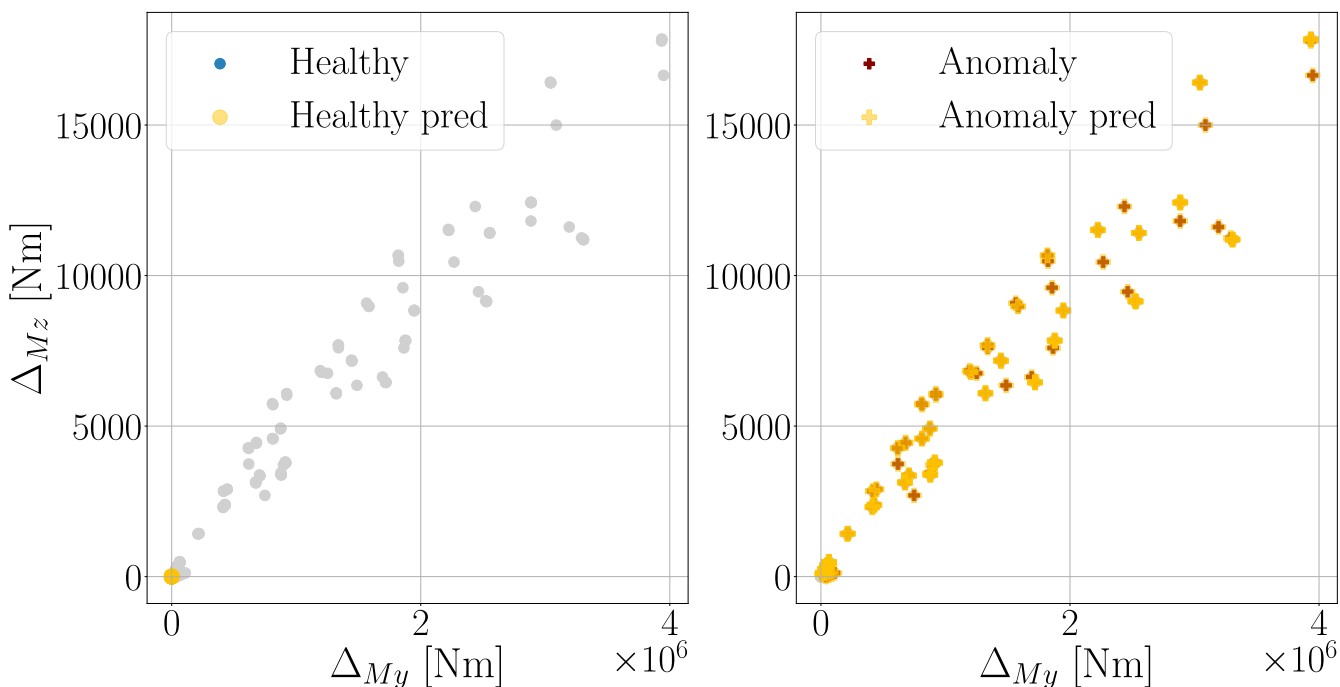

**Figure 4.** Misalignment Detection Assessment Results. This Figure shows a comparison of predicted healthy cases or anomalous cases (in yellow) with the actual healthy and anomaly ones, demonstrating consistency.

| Classification Report | | | | |
|---|---|---|---|---|
| **Label** | **Precision** | **Recall** | **F1-score** | **Support** |
| Healthy | 1.00 | 0.89 | 1.00 | 24 |
| 0.5deg | 1.00 | 1.00 | 1.00 | 23 |
| 1.0deg | 0.92 | 1.00 | 0.96 | 22 |
| 1.5deg | 0.95 | 0.95 | 0.95 | 19 |
| 2.0deg | 1.00 | 0.93 | 0.96 | 28 |

**Table 2.** Metrics computed from the fault detection and quantification output considering NWP cases



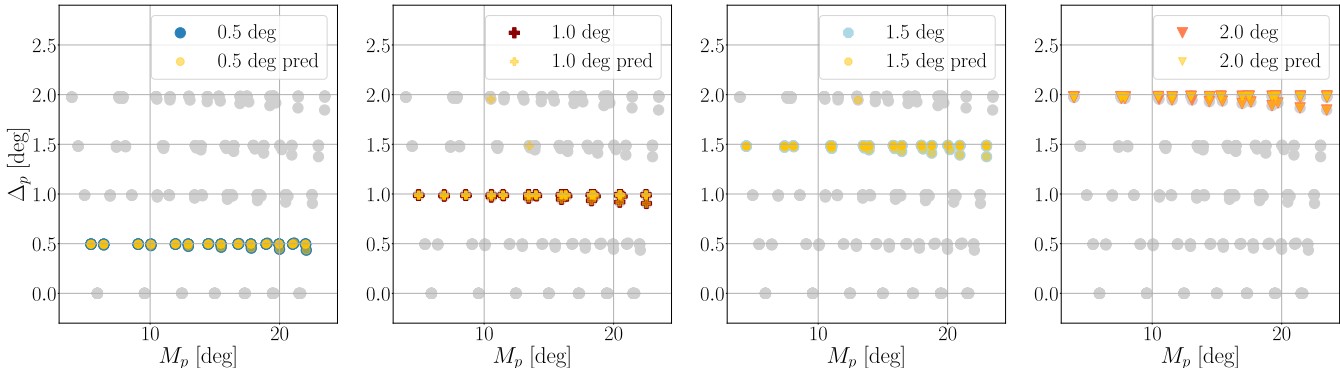

**Figure 5.** Misalignment Severity Assessment Results. This Figure shows a comparison of predicted misalignment severity (in yellow) with the actual one, demonstrating consistency across severity levels.

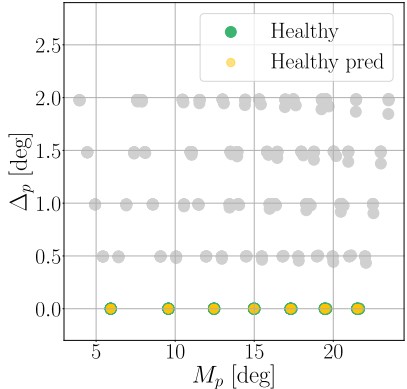

**Figure 6.** Misalignment Severity Assessment Results. This Figure shows a comparison of predicted healthy cases (in yellow) with the actual healthy ones, demonstrating consistency.

starting from a wind speed of $v = 13m/s$ the algorithm is not only capable of detecting the presence of a uniform misalignment
but also to precisely assess its severity. To provide further insights into the classification outcomes, Figure 6 and 5 provide a
graphical representation of the results, showing the actual and the estimated severity degree for each predicted window. In the
plot, each point corresponds to a window, and its coordinates are the mean demanded collective pitch and the delta pitch with
respect to a healthy case. It can be noticed that, due to the effective features engineering, the different misalignment conditions
are well separated, easing the classification task. Indeed, predicted points consistently overlap with actual ones, especially for
the healthy cases. Indeed, given that the system in this scenario is not affected by noise, the $\Delta_p$ are strictly set to 0 leading
to excellent detection capabilities. Also, it can be noticed that the prediction of the model indicates a strong correspondence
between the real and predicted system behavior with small variance with respect to the actual misalignment degree.



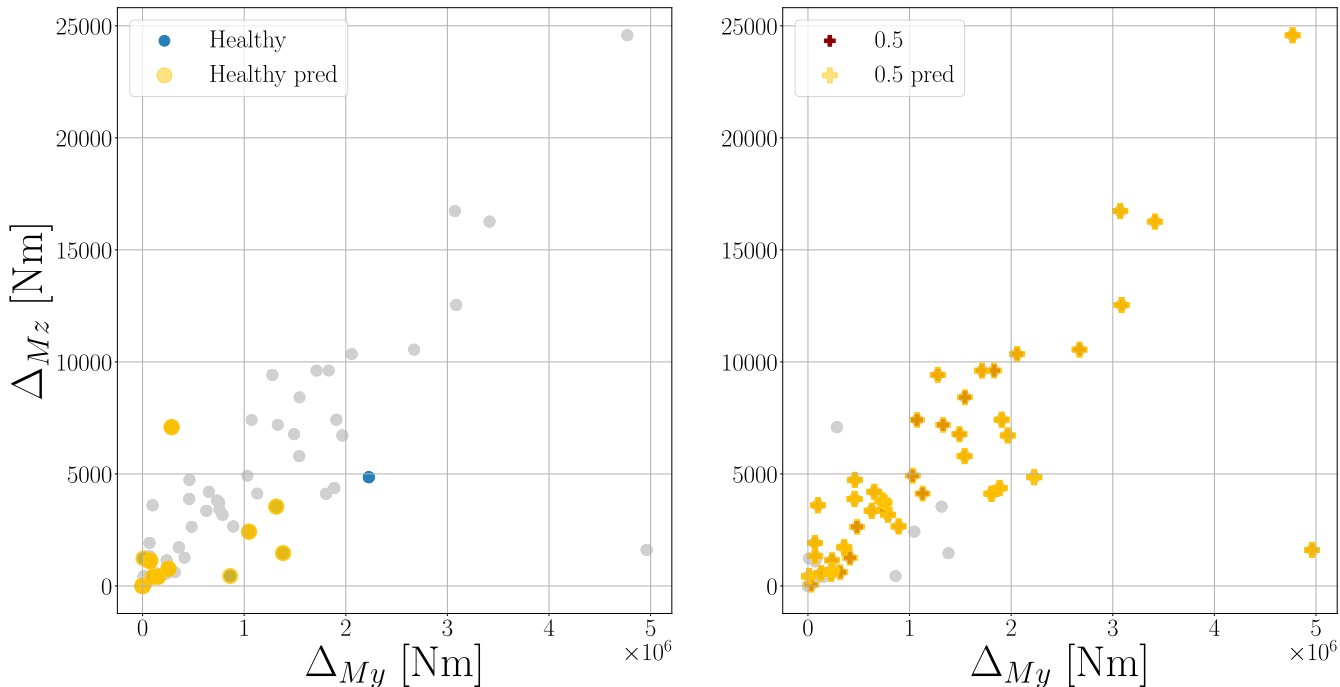

**Figure 7.** Misalignment Detection Assessment Results. This Figure shows a comparison of predicted healthy cases or anomalous cases (in yellow) with the actual healthy and anomaly ones, demonstrating consistency.

## 5.2 NTM data

As previously mentioned, more challenging but more likely operating conditions are still considered in our work, including turbulent scenarios and strong winds. The only limitation is on the minimum wind speed, that increases from $13$ to $15m/s$ for a preciser misalignment quantification. This is due to the fact that, the machine operates across two different power regions, separated by the rated speed as previously mentioned. Given the more turbulent scenario and the presence of noise, distinguishing between the two regions becomes more challenging. Consequently, higher wind speeds are required to clearly identify the operating region and observe the corresponding pitch changes.

### 5.2.1 Region II - below rated speed

As previously done in the stationary case, for a lower wind speed range, the mechanical moments on the blades can be exploited for the anomaly detection. In this case, 10-minute window length is required for a reliable detection. Again, Figure 7 depicts the actual and predicted instances for a wind speed ranging from $v = 5m/s$ to $v = 13m/s$ with an F1-score of $95.71\%$ when considering a model architecture composed of 20 trees with depth 5.



| Classification Report | | | | |
|---|---|---|---|---|
| **Label** | **Precision** | **Recall** | **F1-score** | **Support** |
| Healthy | 0.84 | 0.89 | 0.89 | 24 |
| Misaligned | 0.94 | 0.92 | 0.96 | 48 |

**Table 3.** Metrics computed from the fault detection output considering NTM cases

### 5.2.2 Region III - above rated speed

Conversely, when considering wind speeds higher than the rated one $v = 15m/s$, different scenarios can be explored. We first tried to keep the same window size of the stationary case, namely the short 4.3-minute interval. However, in power region III, this window length only allows for accurate detection of the presence of a uniform misalignment, without enabling a precise quantification of its severity. To achieve such a detailed assessment of the misalignment for higher wind speeds, a larger time window is required. This highlights a trade-off between speed and precision: a rapid response enables only binary detection, whereas a more accurate evaluation of the misalignment severity demands a longer time frame. Given that this specific anomaly evolves over time, and in view of practical applications, a 10-minute window is recommended. This choice also allows the use of the two adopted models developed in the ideal scenario. To begin with, the results for evaluation metrics, including precision, recall, F1-Score, and support are reported in Tab.3 for the detection performance within the short time window of 4.3 minutes. Using the F1-Score as a reference metric, our approach proved effective at detecting the presence of a uniform misalignment, with a promising average F1-Score of $91.5\%$. This result, while slightly lower than in the case of linear wind conditions, still confirms the method's effectiveness in power region III. As previously mentioned, the wind speed in this scenario ranges from 15 to 25m/s. A sensitivity analysis has been carried out by considering different window lengths and the performance metrics for different window sizes are inherently increasing by increasing the windows size. In Figure 8, the F1-score performance is reported when considering three different moving windows. When considering a larger time window, detection performance increase an this lead us to consider a larger time window of 10 minutes for the classification. Thus, with a higher time window, as anticipated, not only the detection of the misalignment can be performed, but a preciser quantification of the entity of the fault can be performed as well. Figure 9 provides a graphical representation of these binary detection results, showing the actual and the estimated misalignment for each predicted window. As before, in the plot, each point corresponds to a window, and its coordinates are the mean demanded collective pitch and the delta pitch with respect to a healthy case. As demonstrated in the Figure, the data are accurately classified despite being more spread in the space. When extending the time window to 10 minutes, not only can higher detection performance be achieved, but a more accurate assessment of the misalignment severity also becomes possible. In Tab.4, the metrics for the quantification procedure performance are reported. Indeed, these achievements stand out as remarkable, since starting from a wind speed of $v = 15m/s$ the algorithm is not only capable of detecting the presence of a uniform misalignment but also precisely assess its severity with an average F1-score of 96.82%. In this case the model architecture becomes slightly more complex encompassing 30 trees with depth 5. To provide further



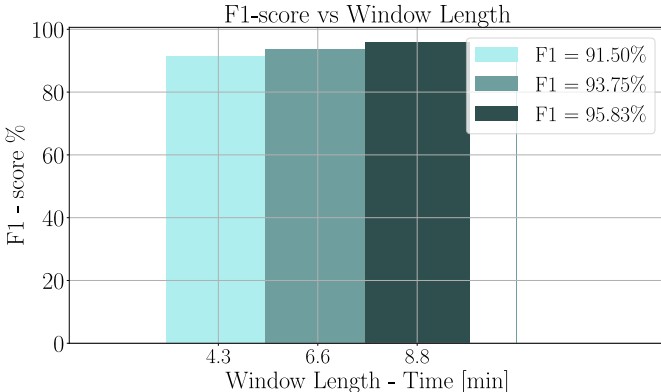

**Figure 8.** Time window performance sensitivity. This bar plot shows how performance change across different time window lentgh: the longer the time window the higher the performance.

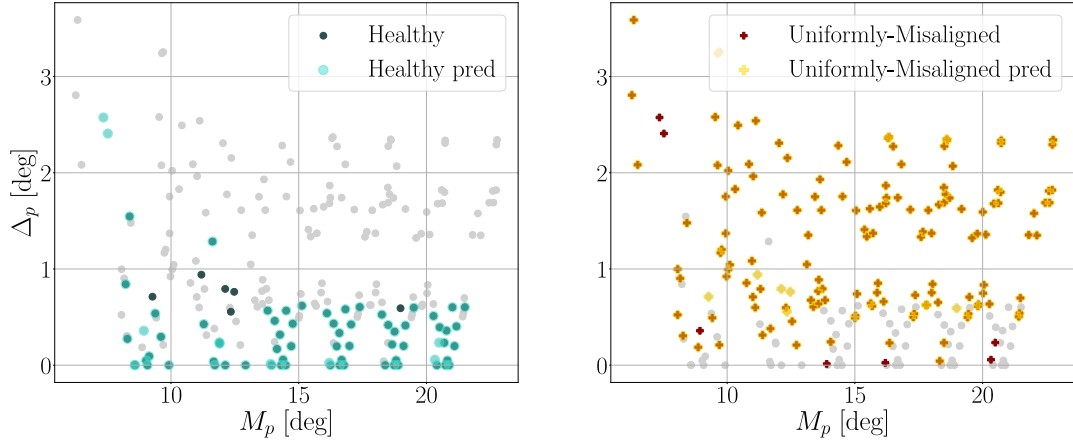

**Figure 9.** Misalignment Detection Results. This Figure shows a comparison of predicted misalignment in lighter colors with respect to the actual one in darker color, demonstrating consistency across different wind speeds.

| Classification Report | | | | |
|---|---|---|---|---|
| **Label** | **Precision** | **Recall** | **F1-score** | **Support** |
| Healthy | 0.9 | 0.79 | 0.78 | 12 |
| 0.5deg | 1.00 | 1.00 | 1.00 | 11 |
| 1.0deg | 1.00 | 0.83 | 0.91 | 11 |
| 1.5deg | 0.85 | 1.00 | 0.86 | 10 |
| 2.0deg | 0.8 | 0.91 | 0.88 | 14 |

**Table 4.** Metrics computed from the fault detection and quantification output considering NTM cases





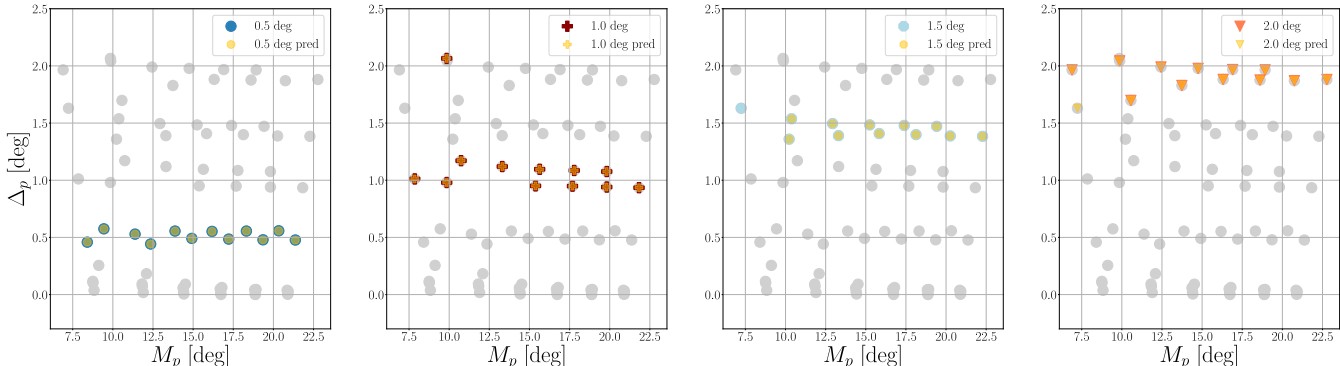

**Figure 10.** Misalignment Severity Assessment Results. This Figure shows a comparison of predicted misalignment severity (in yellow) with the actual one, demonstrating consistency across severity levels for the turbulent wind scenario.

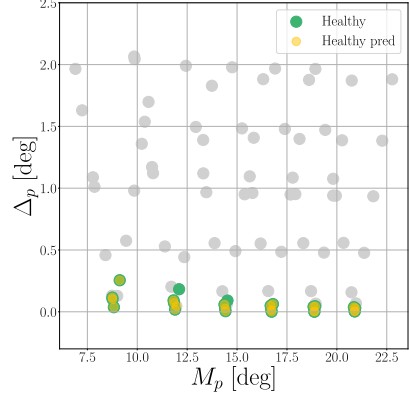

**Figure 11.** Misalignment Severity Assessment Results. This Figure shows a comparison of predicted healthy cases (in yellow) with the actual healthy ones, demonstrating consistency for the turbulent wind scenario.

insights into the classification outcomes, Figure 10 and 11 provide a graphical representation of these results, showing the actual and the estimated severity degree for each predicted window. As remarked, the prediction of the model indicates a strong correspondence between the real and predicted system behavior, although the variance and spread of points is higher with respect to the ideal case without turbulence.

## 6 Conclusions

In this work, we build upon our previously developed method machine-learning framework specifically tailored for detecting a-symmetrical distributed pitch misalignment in wind turbines. Our goal was to analyze the challenging condition where all the blades are uniformly-misaligned by the same degree. In fact, this condition, despite being critical, has been scarcely explored in the literature given its complexity and the overlapping behavior of this case with a healthy scenario. Considering the presented





results, the proposed approach proves robustness in both ideal and turbulent wind scenarios and different wind speeds, while preserving its inherent interpretability. These remarkable results have been achieved by leveraging ad-hoc features that are related to the physical behavior of the system. More in detail, our strategy extracts time domain features encompassing the behavior of the required collective pitch demanded to the blades, which we addressed after an accurate sensitivity analysis as

the primary indicator of the fault above the rated wind speed. While, for lower wind speeds, we relied on the time domain behaviour of the local mechanical moments on the blades. The mean and standard deviation of these signal and the deviation with respect to a non-anomalous case have been exploited for detecting the presence of rotor imbalances and to assess the severity of the misalignment. The validation of this approach has been performed through simulations featuring non and turbulent wind conditions, different wind speeds and pitch misalignment degrees. This procedure allowed us to asses the

performance of the method in detecting and precisely quantifying this anomaly that leads to symmetrical behavior of the system that can mask the fault. More specifically, the method yields an average F1-score up to $97.41\%$ in detection and classifying the anomaly in case of an ideal scenario ranging from a pitch misalignment from $0.5$ to $2.0$ deg on all blades encompassing a wind speeds ranging from $5$ to $25 m/s$. When considering the more realistic yet noisier scenario of turbulent wind, larger time is required to have optimal performance. Still, the framework achieves an average F1-score of $96.82\%$ in the

detection and quantification of this anomaly requiring a time window of 10 minutes. Thus, this work proves to be effective at filling the missing gap in the literature with a more comprehensive and precise fault diagnosis, that is capable to address also this peculiar fault scenario. These advancements might support the shift from a time-based to a condition-based approach for turbine maintenance scheduling, minimizing downtime and maximizing energy production.

*Author contributions.*  All authors provided fundamental inputs to this work through discussions, feedback, and analyses of the obtained

results. S.M., J.L. and M.T. devised the main idea underlining the machine learning-based detector. S.C. and A.C. identified the physical principles the detector algorithm is anchored on. S.C. and A.C. performed the multibody simulations of the wind turbine in healthy and unhealthy conditions. S.M., J.L. and M.T. implemented and performed the detection strategy.

*Competing interests.*  AC is member of the editorial board of *Wind Energy Science*.



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
