# Peer review of "Uniform Blade Pitch Misalignment in Wind Turbines: a learning-based detection and classification approach"

_Wind Energy Science, 2025_

## Referee Comment (RC1)

**Comments to wes-2025-153**

**General comments**

The paper proposed a data-driven method for detecting collective turbine pitch offsets. The topic is relevant to wind energy, and the methods are well described. Some minor changes to the manuscript are advised in order to improve its impact and readability.

The literature review is a bit too concise, and the authors should spend some time finding more references for the topic (at least 10). The topic of wind turbine proactive monitoring is quite vast, and it feels that many important works were not mentioned.

The titles of sections 2 and 3 are similar, which may disorient the reader. "Preliminary" and "Exploratory" are also repeated. The paper would benefit from a restructuring into a more common layout including only Introduction, Methods, Results, and Discussion.

A point that was skimmed over is the definition of the inflow. In particular, the turbulent class is just briefly mentioned, but it would be important to understand wind conditions and terrain type the inflow turbulence represents.

The accuracy of each of the 4 models should be reported in a more consistent way. All sections should have a table reporting recall, precision, F1 score, and support, and possibly also a confusion matrix (particularly when using the binary detection, like in Region II). Also, the existing tables and figures are missing important information in their captions, so it is hard to pinpoint the data they refer to.

To conclude, it would be interesting to discuss the real-world challenges that implementing this method on a real turbine would inevitably bring about. Think, for instance, about how you would train a model for full-scale application (using SCADA data or still simulations?), how different turbulence conditions may affect accuracy, if industry cares more about false negatives (i.e., missed alarms) or false positives (i.e., unnecessary inspections), etc.

**Specific comments**

L91: Should "Normal Weather Prediction" be "Numerical Weather Prediction". This is what NWP stands for, generally. Please add a reference for this as well.

L106: Please define *1xRev* and make its formatting consistent.

L121: It is not clear why the pitch regulator would still maximize power in the case of a pitch misalignment. It sounds like the pitch offset, as defined, is indeed the difference between

the pitch command and the actual pitch, so it should lead to suboptimal performance in Region II.

L130: "fro" instead of "from".

L174: There is a "1" in the text.

L176: Missing parenthesis for citation.

L186: It is recommended to turn the definition of the metrics into a table or formulas that use the more common definitions of False/True positives and False/True Negative (see e.g., https://www.osti.gov/biblio/1769877)

Fig. 4: These plots are a bit unclear:

- Are gray dots all the cases?
- Where are the blue dots on the left panel?
- It is unclear when anomaly and anomaly predictions superimpose. It would be better to use a cross and an empty circle as markers or play with transparency, for example.

L215: "model'" is a typo.

L261: The concept that a 10-minute window leads to better classification performance was already introduced at L256, consider removing it.

---

## Referee Comment (RC2)

[referee-annotated manuscript omitted]